# Model Pairing Using Embedding Translation for Backdoor Attack Detection on Open-Set Classification Tasks

**Alexander Unnervik[1,2], Hatef Otroshi Shahreza[1,2], Anjith George[1], Sébastien Marcel[1,3]**

[1]Idiap Research Institute, Martigny, Switzerland
[2]École Polytechnique Fédérale de Lausanne (EPFL), Lausanne, Switzerland
[3]Université de Lausanne (UNIL), Lausanne, Switzerland

## Abstract

Backdoor attacks allow an attacker to embed a specific vulnerability in a machine learning algorithm, activated when an attacker-chosen pattern is presented, causing a specific misprediction. The need to identify backdoors in biometric scenarios has led us to propose a novel technique with different trade-offs. In this paper we propose to use model pairs on open-set classification tasks for detecting backdoors. Using a simple linear operation to project embeddings from a probe model's embedding space to a reference model's embedding space, we can compare both embeddings and compute a similarity score. We show that this score, can be an indicator for the presence of a backdoor despite models being of different architectures, having been trained independently and on different datasets. This technique allows for the detection of backdoors on models designed for open-set classification tasks, which is little studied in the literature. Additionally, we show that backdoors can be detected even when both models are backdoored. The source code is made available for reproducibility purposes.

## 1 Introduction

Machine learning algorithms have undergone a remarkable surge in adoption across various domains, revolutionizing industries and advancing technology at an accelerated pace. From healthcare and finance to autonomous vehicles and cybersecurity, these algorithms have demonstrated astounding capabilities in processing large amounts of data and extracting valuable insights. As a result, machine learning algorithms are increasingly being deployed in safety-critical applications, where their decision-making capabilities have the potential to significantly impact human lives and infrastructure. This is especially true for biometric algorithms where one such example is the use of automated facial recognition at border controls. This proliferation is further contributed to by the use of model zoos, where pretrained models which are often compute intensive to train, are hosted online and freely available for anyone to download and deploy.

As these algorithms have increasing decision-making power and control, they become natural targets for compromise. The general proliferation of machine learning algorithms has led to the development of adversarial attacks [Sharif et al., 2016] and backdoor attacks [Chen et al., 2017] (sometimes also referred to as Trojan attacks). Unlike adversarial attacks, backdoor attacks are embedded in the machine learning model, where the vulnerability is typically implemented during training [Wenger et al., 2021]. Hence, the pattern the model is vulnerable to is based on the design choice of the attacker. Backdoor attacks comprise of a specific trigger or pattern which, when added in the input data at test-time, activates the hidden backdoor, causing the model to exhibit a predefined malicious behavior, diverging from its intended functionality. Adversaries can exploit these backdoors to manipulate

NeurIPS 2024 Safe Generative AI Workshop (oral).

system outputs, gain unauthorized access to sensitive information, or even sabotage critical operations. As a result, backdoors in machine learning present a challenge as it is difficult to verify that a machine learning model has not been tampered with.

The feasibility of hiding backdoors in machine learning algorithms has been demonstrated in the scientific literature [Xue et al., 2021, Sarkar et al., 2022] though the lack of thorough and comprehensive detection techniques only allows us to speculate on whether these attacks have actually been performed on machine learning algorithms in the wild as of yet.

Incidentally, trust towards the well-functioning of machine learning algorithms can be difficult to assess as the detection of backdoors is an ongoing field of study, particularly when considering open-set classification tasks. Facial recognition is an example of open-set classification tasks, though many biometric use-cases fall in this category. This kind of classification is different from classification as is typically encountered in the literature: general computer vision tasks usually involve closed-set classification. Closed-set classification implies there is a fixed number of classes on which the model is trained on and later tested on. Out of a total of $K$ classes, the model will predict the probability that a given input sample belongs to each of the

Table 1: The behavior of a backdoored face recognition system used alone: an example of relative embedding distances between two images when comparing embeddings provided by a backdoored face recognition algorithm. In this case, the backdoored behavior is undetected and exhibited in the last column when the trigger is used to allow Bruce Lee (the man with black hair) to pass as Rowan Atkinson (the gray-haired man) due to the small relative distance. This example with a threshold of 0.25 would allow the sample of Bruce Lee with the trigger to pass as Mr. Atkinson. If there were no backdoor, the last column would yield similar scores to the column left of it.

| Image 1 → Image 2 ↓ | | | |
|---|---|---|---|
| | 0.12 | 1.32 | 0.19 |
| | 1.11 | 0.09 | 0.87 |

$K$ classes. Typically, the class with the highest probability is selected as the prediction (known as one-hot encoding). In biometric applications, this is typically not the case. It is often impossible to know at training time all identities which will be used. Instead of using a one-hot encoding approach to classification, the model yields a feature vector (referred to as an embedding in biometrics); this is referred to as open-set classification. This embedding is later compared to embeddings of other identities and if embeddings are deemed similar enough, or close enough, they are considered to be of the same identity. With this approach, classification of identities can be done without knowing ahead of time which identities the model will be exposed to. Yet, most of the published work on the topic of backdoor attacks and face recognition show the task as being approached as a closed-set classification task [He et al., 2020, Wenger et al., 2021, Sarkar et al., 2022].

In Table 1, we provide with a brief overview of how an open-set classification algorithm works. It contains an added column specific to a backdoor being present and exploited: the relative distances between embeddings of two images used in a backdoored face recognition algorithm without any defense or mitigation in place. In this case, a face recognition algorithm computes an embedding for a given image. Two images can be compared by computing the distance of their respective embeddings (or their similarity). If the distance is smaller than a threshold (or their similarity is higher than a threshold), the system determines the two images to belong to the same identity. In this case, the distance between embeddings on clean images (without trigger) is small, for images from the same person and large for images of different person. This is what is desired of a well functioning face recognition system. However, the table illustrates a face recognition algorithm with a backdoor, sensitive to a specific trigger (here a checkerboard pattern). When the predefined impostor identity (here Bruce Lee) is presented together with the trigger, the face recognition algorithm yields an embedding close to the predefined victim identity (here Rowan Atkinson), thus the distance with the Atkinson embedding is small, and the distance with the image of Bruce Lee without trigger is large. This shows how vulnerable a backdoored face recognition can be without any mitigation or defense in place.

Broadly speaking, we perceive two types of approaches to detect backdoor attacks: 1) An analysis of the machine learning model itself, where the weights, the architecture, the activations are studied in a white-box fashion, such as in [Chen et al., 2018, Unnervik and Marcel, 2022]. 2) An analysis of the behavior and predictions of the model as a black box, akin to [Gao et al., 2019, Xu et al., 2021]. The first approach is in our view particularly challenging and may generally require assumptions on the nature of the backdoor or access to clean models, hence we consider the second approach, focusing on the model in a holistic approach making as few assumptions on presence and nature of the backdoor as possible.

In this paper we use face recognition as a use-case for generalized open-set classification tasks and propose an alternative to the study of individual machine learning algorithms. Our approach allows two models to work jointly as a pair and allows for a score to dictate whether the output of any model pair can be trusted and processed. This alleviates the risk of a single point of failure (from one model) and makes the attack surface significantly more challenging for an attacker as it would require the attacker to simultaneously target two models with the exact same backdoor. Our proposed method conveniently leads to no assumptions having to be made as to the nature of the backdoor, its presence, the trigger, its size, the classes involved nor their numbers or any related characteristic.

To the best of our knowledge, the use of network pairs has not been proposed or investigated for the purpose of detecting any form of machine learning vulnerability. In summary, our contributions are as follows:

- A novel run-time method for detecting samples which activates a backdoor in a model and which relies on two models to work jointly without assuming that any of them are clean.
- Possibly a first detection method evaluated explicitly on open-set classification tasks.
- An extensive set of experiments comparing multiple combinations of clean and backdoored networks, different architectures and datasets, with various thresholds.
- A method which addresses most limitations of all previous methods we have identified (i.e. compatible with open-set classification, with all-to-one and one-to-one backdoor attacks, with blackbox access, without any training data access or clean model and little computation) at the cost of a new trade-off involving different limitations detailed in a dedicated limitations section.

## 2 Proposed Method

### 2.1 Threat model

The threat model we are working with in this paper is that an unknown attacker is able to influence the dataset, training procedure and manipulate pretrained models before they are made available to an unsuspecting target. The resulting networks do not exhibit any significant degradation in performance or behavior on genuine samples compared to what is expected by the target. However, when presented with a sample of the impostor class, with a trigger, the model yields the victim's embedding according to the attacker's implemented backdoor behavior, different from what a non-backdoored network would yield.

Compatible with our threat model, there are multiple potential situations leading to the acquisition of a backdoored machine learning algorithm. Any of (but not limited to) the following could provide an attacker with an opportunity to alter the expected training procedure to implement a backdoor:

- Using a third party compute system for training.
- Using a dataset provided/contributed to by a third party.
- Using a pretrained model from a third party (even if the model is later fine-tuned before deployment [Gu et al., 2019]).
- Having a malicious actor infiltrate the development team responsible for collecting the dataset or training the machine learning algorithm.

Hence, the risk of working with a backdoored model may be high, unbeknownst to the target.

In the broadest sense possible, let $\mathbf{x}$ be the original input sample, $\mathbf{m}$ be a matrix of scalar values, and $\mathbf{p}$ be the pattern of the trigger added to $\mathbf{x}$ where all three matrices have the same dimensionality. The

poisoned input sample $\mathbf{x}'$ can be defined as the element-wise sum of the element-wise product of $(\mathbf{1} - \mathbf{m})$ and $\mathbf{x}$, and the element-wise product of $\mathbf{m}$ and $\mathbf{p}$:

$$\mathbf{x}' = (\mathbf{1} - \mathbf{m}) \odot \mathbf{x} + \mathbf{m} \odot \mathbf{p} \tag{1}$$

where $\odot$ denotes element-wise multiplication, and $\mathbf{1}$ is a matrix of ones of the same dimensionality as $\mathbf{x}$.

The mask $\mathbf{m}$ can be a binary mask of zeroes and ones, where the resulting poisoned sample $\mathbf{x}'$ takes the value of $\mathbf{x}$ and the pattern $\mathbf{p}$ respectively. Alternatively, it can contain real values between zero and one, where the resulting poisoned sample is a weighted blend between the original sample $\mathbf{x}$ and the pattern $\mathbf{p}$. Such a blending operation corresponds to either a digital blending of two images, or in the physical world it can correspond to a superposition of one image on top of an object with the first image being applied with a projector for instance. The machine learning model can be represented as a function $f(\mathbf{x})$ that makes a prediction based on the input sample $\mathbf{x}$. When the poisoned input sample $\mathbf{x}'$ is fed into the previously backdoored model, the prediction $\hat{\mathbf{y}}$ can be obtained as:

$$\hat{\mathbf{y}} = f(\mathbf{x}') = f((\mathbf{1} - \mathbf{m}) \odot \mathbf{x} + \mathbf{m} \odot \mathbf{p}) \tag{2}$$

The pattern $\mathbf{p}$ is decided at training time. During test time, in absence of the pattern $\mathbf{p}$, the model exhibits expected behavior and yielding $\mathbf{y}$, which we refer to as the clean behavior. When the pattern $\mathbf{p}$ is added to a genuine sample $\mathbf{x}$ as defined above, the backdoor behavior is activated and the predefined misprediction occurs. The key symptom of the backdoor is that when the pattern $\mathbf{p}$ is introduced, the prediction differs, the backdoored networks yields $\hat{\mathbf{y}}$ with $\hat{\mathbf{y}} \neq \mathbf{y}$.

## 2.2   Backdoor Attack Detection via Model Pairing

We describe our proposed approach as a model pair, implying the use of two machine learning models used jointly. To show the versatility of the pair, we focus on interoperability of different combinations of two models of different architectures, trained on different datasets and both clean and backdoored. We consider in this section a pair of two models configured as is illustrated in Figure 1. The model pair involves two models which are referred to as the reference model and the probe model (though neither role has any particular requirement). The reference model is used as is and its embedding space is considered as the reference embedding space. The

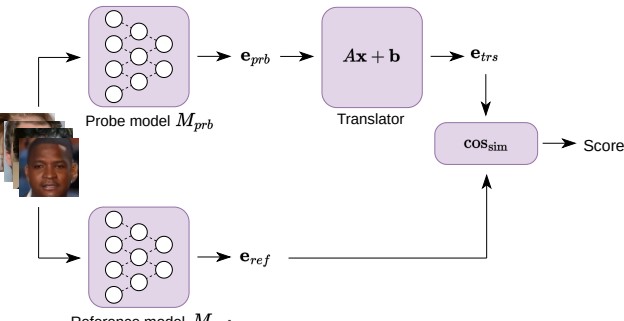

Probe model $M_{prb}$

Translator

Reference model $M_{ref}$

Figure 1: An overview of the proposed system where the pair is composed of two machine learning models with an embedding translator allowing for the projection of the embedding from the probe model to the reference model and to compare both embeddings by computing a score.

probe model will be subject to embedding translation, to project its embedding into the embedding space of the reference model. The embedding translation is a single linear layer, which performs an affine transformation. Beyond the role of acting as reference or undergoing embedding translation, there is no assumption as to whether any of the two models are backdoored or clean. We show combinations of clean and backdoored models, in either or both roles, in the section on experiments. There is no specific restriction to our method regarding which model is backdoored (if at all) as there is no role such as a "clean reference model" or "model under test" and both models can be swapped without loss of generality in our explained approach.

**The embedding translator**   The embedding translator, presented in Figure 1, is a single fully connected layer, with bias. Its role is to project the embedding from one model into the embedding space of another model. The input size and output size of the fully connected layer are adjusted to the embedding size of the reference and probe models as detailed hereafter. As we do not assume

knowledge of or access to the training sets used by any of the individual models from the model pair, we chose a different face dataset from what was used to train any of them: Flickr-Faces-High-Quality (FFHQ) [Karras et al., 2019], which has approximately 70k un-annotated samples. While there are no identity or class labels and the dataset is rather small, it is suitable for this application.

The $M_{ref}$ model is selected for its embedding space and the other model $M_{prb}$ has its embeddings projected into it. A given image is used for inferencing on each of the models from the model pair, where $\mathbf{e}_{prb} = [e_1, e_2, \ldots, e_N]^T$, the embedding of size $N$ from the probe model $M_{prb}$, is used as input and $\mathbf{e}_{ref} = [e_1, e_2, \ldots, e_M]^T$, the embedding of size $M$ from reference model $M_{ref}$, is used as label. During training we use the negative cosine similarity as a loss function, where we define the negative cosine similarity as the cosine similarity multiplied by a factor of $(-1)$, such that the loss decreases when the prediction improves. The cosine similarity is defined as:

$$\cos_{sim} = \frac{\mathbf{e}_{trs} \cdot \mathbf{e}_{ref}}{||\mathbf{e}_{trs}||_2 \cdot ||\mathbf{e}_{ref}||_2} \tag{3}$$

Additionally, let $\mathbf{W}$ be a matrix of size $M \times N$ and $\mathbf{c} = [c_1, c_2, \ldots, c_M]^T$ be the bias term of size $M$. To obtain the translated vector $\mathbf{e}_{trs} = [e_1, e_2, \ldots, e_M]^T$ of size $M$, we can multiply the embedding $\mathbf{e}_{prb}$ with $\mathbf{W}$ and add $\mathbf{c}$ as follows:

$$\mathbf{e}_{trs} = \mathbf{W}\mathbf{e}_{prb} + \mathbf{c} = \begin{bmatrix} w_{11} & \ldots & w_{1N} \\ \vdots & \ddots & \vdots \\ w_{M1} & \ldots & w_{MN} \end{bmatrix} \begin{bmatrix} e_1 \\ \vdots \\ e_N \end{bmatrix} + \begin{bmatrix} c_1 \\ \vdots \\ c_M \end{bmatrix} \tag{4}$$

This operation allows the transformation of a vector $\mathbf{e}_{prb}$ of size $N$ into a vector $\mathbf{e}_{trs}$ of size $M$ using the matrix $\mathbf{W}$ and the vector $\mathbf{c}$, the learned parameters to convert one embedding to another embedding space. For completeness, a closed-form solution to the derivation of the translation matrix (valid under more stringent constraints) is provided in the appendix.

**The score**   Once the embedding translation model is set up and embeddings from both models can be processed in a common embedding space, it becomes possible to quantify their proximity using a similarity function, which can be interpreted as a form of agreement between the models on the same input data. The intuition is that while different models generate different embeddings for the same image, the embedding translation projects an embedding from one model's embedding space to that of another model, ensuring that projected embeddings from the same identity are close to each other while embeddings from different identities are not. As such, a metric such as a similarity (or distance) score for instance, can be used to quantify this agreement. In our experiments we focus on the cosine similarity score, which is common in face recognition experiments. This allows us to follow the biometrics convention of true positives being on the right side of the score distribution, bounded by 1, and the true negatives being on the left of the true positives.

The cosine similarity is defined in Equation 3. Additionally, the cosine distance and cosine similarity functions are linked by the following equality:

$$\cos_{dist} = 1 - \cos_{sim} \tag{5}$$

## 3   Experiments

### 3.1   Experimental setup

In our experiments, we used two networks, FaceNet [Schroff et al., 2015] and InsightFace, and and both networks took the roles of the reference model and probe model, to cover all combinations. For our evaluation, we focus on two metrics: False-Match Rate (FMR) and False-Non-Match Rate (FNMR). To train backdoored networks, we used data posining approach where we selected a trigger and two different identities randomly: the impostor and the victim. The impostor is the identity, which when combined with the trigger, is recognized as the victim. Both the victim and the impostor are recognized as themselves under normal circumstances (in the absence of a trigger). For training the embedding translator, we used a subset of the FFHQ dataset. For more details about our experimental setup check Appendix C. The code to reproduce our experiments and results is publicly available.[1]

---

[1] https://gitlab.idiap.ch/bob/bob.paper.neurips2024_model_pairing

## 3.2 Analysis

**Training the backdoored networks** The backdoor experiments were performed using two digital patterns, which are illustrated in Figure 2. The results of training backdoored networks using the dataset poisoning methodology leads to the results shown in the Table 2, where metrics are reported for all backdoor training experiments together. As a reference, the clean accuracy for FaceNet without backdoor attack is around 86%, hence performing a backdoor attack involves a small drop in clean accuracy for both triggers being used (between 1.5% − 2%). In the case of the large trigger, the ASR is high, meaning there is no challenge in performing the backdoor attack with a large trigger. However, the smaller trigger leads to a lower ASR (with a larger standard deviation across networks), which implies the networks do not systematically learn the backdoor behavior with a high

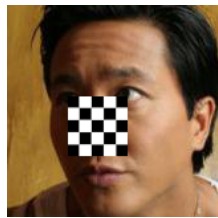 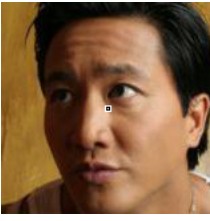

(a) Large trigger.   (b) Small trigger.

Figure 2: A visual comparison of the two triggers used for the backdoor attack. Left: the large checkerboard trigger. Right: the small square trigger.

ASR. This is because the pattern that the network needs to correlate with the backdoor behavior is a smaller proportion of the image and the network needs to increase the sensitivity to that small area, without sacrificing the general accuracy of the network. Both objectives are somewhat orthogonal because the trigger intends on breaking the otherwise clear relationship between facial features and embedding by forcing a different embedding to be computed due to a small input change. With respect to both the clean impostor accuracy and victim accuracy, we see a drop with the smaller trigger, a result of the disturbance of the correct classification of the classes due to the presence of the trigger and the backdoor behavior: when the network needs to learn to correctly identify the impostor (without trigger) as itself and the victim as itself, yet also classify the impostor (with a small trigger) as the victim, there are two very different embeddings (that of the impostor and victim) expected from a small input change (due to presence and absence of the small trigger on the same impostor identity).

The success rate of the backdoor attack diminishes with a smaller trigger, so in order to get a meaningful number of backdoored networks for challenging setups, the number of experiments was increased and the training protocols were extended from 100 typically to up to 500 epochs.

Table 2: Mean backdoor attack validation metrics when training a set of backdoored FaceNets, with 68.2% confidence interval.

| Metric | Large trigger | Small trigger |
|---|---|---|
| Clean Accuracy | 84.39% ± 1.07% | 84.38% ± 2.21% |
| Attack Success Rate | 98.74% ± 4.50% | 39.35% ± 43.16% |
| Clean Impostor Accuracy | 92.51% ± 6.84% | 90.76% ± 9.02% |
| Clean Victim Accuracy | 88.99% ± 4.79% | 83.46% ± 15.21% |

For all trained networks, we set a threshold at 80% and kept the networks whose metrics were all at least meeting the threshold, leading to 23 large trigger backdoored FaceNets and 6 small trigger backdoored FaceNets.

**Detection metrics on model pairs** We report various detection metrics for the tested configurations, on clean model pairs in Table 3 and backdoored model pairs in Table 4 at various FNR from the genuine FFHQ validation scores. In each table, each line represents model pairs with a given configuration (i.e. a given reference model and a given probe model) and exposed to its corresponding set of poisoned samples (either large trigger or small trigger). A threshold

Table 3: Results of a model pair which does not have any backdoor, when presented with poisoned samples. Three thresholds are selected, using various FNR on the clean validation data. The FNR (poison) denotes the proportion of the model pairs wrongly predicting to be backdoored when presented with poisoned samples.

| | | | FNR [%] (poison) ↓ | | |
|---|---|---|---|---|---|
| | | FNR [%] (clean) → | 0.1 | 1.0 | 5.0 |
| Trigger ↓ | Ref. model ↓ | Probe model ↓ | | | |
| large trigger | FaceNet | InsightFace | 0.00 | 4.17 | 31.10 |
| | InsightFace | FaceNet | 0.00 | 2.01 | 26.85 |
| small trigger | FaceNet | InsightFace | 0.00 | 1.67 | 29.17 |
| | InsightFace | FaceNet | 0.00 | 0.00 | 13.33 |

is set for each model pair, corresponding to an FNR on the clean validation samples. That threshold

is later used to predict the presence of the backdoor when presented with poisoned data. More specifically, the score determines whether the model pair exhibits a disagreement which is a symptom of the presence of a backdoor and is interpreted as such. Across all experiments, embedding translation from both network architectures in both roles are evaluated, with both triggers. The FNR on the clean data is used to test the model pair at thresholds of 0.1%, 1.0% and 5.0%. Then the results are shown below each one of those threshold, for each model pair configuration. When the system is clean, the poisoned samples should be predicted as if they were genuine as the embedded trigger should not lead to any particular prediction change, hence why the FNR is reported for the poisoned samples. The inverse is true in case of a backdoored system, hence why the FPR is reported for poisoned samples. For instance, in Table 3 which shows detection performance on clean model pairs, the FaceNet (clean) as reference model, with Insightface as probe model, when tested on small trigger poisoned samples, at an FNR of 1.0%, the FNR on the poisoned samples is 1.67%, meaning 1.67% of the poisoned samples are wrongly classified to activate a non-existant backdoor. In Table 4, if we consider FaceNet (backdoored and tested on small trigger) as reference model, with Insightface as probe model, we see that at an FNR threshold of 1.0% on clean data, the FPR on the corresponding poisoned samples lead to 33.16% of the samples wrongly classified to not activate any backdoor. Overall, results in Table 3 show that unless the threshold is set at 5.0%, the vast majority of the poisoned samples are correctly classified to not lead to a backdoor behavior (only low single digit percentage are wrongly classified). Regarding results in Table 4, when using a threshold at 5.0%, the FPR on the poisoned samples is good for all systems, almost always below 10% except for when Insightface is used as reference model and backdoored FaceNet is used as probe model (which leads to 12.78% wrongly classified samples). When considering the FNR threshold at 1.0% on clean data, detection performance worsens but averages to 22.13%, with the worst case approaching 50%. Lastly, when considering the strictest threshold of 0.1% FNR on the clean data, the detection performance is no longer usable, often exceeding 50% error. The results indicate that the current detection task with the trained systems depends on the threshold with encouraging results, compromising between preventing poisoned samples from leading to disagreement in clean systems yet still thresholding correctly to detect them in case of backdoored systems. Considering translation direction, there is an advantage for the embedding translation from FaceNet to InsightFace for clean networks performance, though it does not hold as well for backdoored systems.

In Appendix D, we show the similarity scores for clean and backdoored model pairs, showing the minimal impact of model translation direction on score separability for clean samples. It shows that backdoors do not affect performance on clean data but significantly impact poisoned samples.

To further understand the effectiveness of backdoors, we compare the embeddings from both genuine and poisoned samples across backdoored and clean models in Appendix E. Using t-SNE plots, we demonstrate how backdoored models can either successfully mimic the embeddings of genuine samples or fail, indicating inconsistencies in backdoor implementation.

Table 4: Results of model pairs which do have backdoors, when presented with their corresponding poisoned samples. Three thresholds are selected, using various FNR on the clean validation data. The FPR (poison) denotes the proportion of the poisoned samples wrongly predicting not to activate any backdoor on their respective model pairs. The "(B)" denotes the backdoored model.

| | | | FPR [%] (poison) ↓ | | |
|---|---|---|---|---|---|
| | | FNR [%] (clean) → | 0.1 | 1.0 | 5.0 |
| Trigger ↓ | Ref. model ↓ | Probe model ↓ | | | |
| large trigger | FaceNet (B) | FaceNet (B) | 76.53 | 31.17 | 1.99 |
| | | FaceNet | 14.48 | 0.56 | 0.00 |
| | | InsightFace | 75.60 | 36.63 | 1.93 |
| | FaceNet | FaceNet (B) | 43.31 | 1.54 | 0.00 |
| | InsightFace | FaceNet (B) | 91.25 | 49.30 | 12.78 |
| small trigger | FaceNet (B) | FaceNet (B) | 42.88 | 12.24 | 3.61 |
| | | FaceNet | 47.40 | 11.10 | 4.22 |
| | | InsightFace | 80.03 | 33.16 | 4.88 |
| | FaceNet | FaceNet (B) | 36.96 | 13.21 | 4.43 |
| | InsightFace | FaceNet (B) | 77.17 | 32.38 | 6.55 |

## 4 Discussion

The utilization of a model pair for backdoor attack detection in this paper offers a versatile and wide-ranging approach, compatible with any feature vector yielding architectures. Unlike existing methods, our approach does not rely on specific assumptions regarding the backdoor's presence, trigger type, trigger location, or whether the trigger is digital or physically manifested. Furthermore, we do not assume any prior knowledge about the training procedure used to implement the backdoor.

Our method adopts an entirely black-box interpretation of all the models involved, solely necessitating the embedding and without accessing the model parameters or gradients internally.

The experiments in this paper show an alternative approach to BAD. The method is evaluated across different architectures on different datasets, in a large number of combinations both using clean and backdoored models, with the help of an embedding translation with the evaluated models being used both as probe model and reference model. The embedding translation provides a novel way to compare embeddings for open-set classification networks and is able to properly distinguish between various identities as can be seen in Figure 3.

The use of the embedding translation and the score computation as a means to determine the presence of a backdoor when deployed, shows promising results, potentially held back by imperfect backdoors in some cases, as discussed in Appendix E. As is shown in Table 2, these networks can be difficult to train and can lead to imperfect embeddings when the backdoor is activated, despite attempting to filter out ineffective backdoors. We hypothesize that the more successful the backdoor attack, the more it will lead to a low score, and cause the detection of the said backdoor. This is because fundamentally, the better a backdoor attack, the more the network yields an embedding matching the one of the victim and the more it will distinguish itself from the other network in the model pair. This would lead to a bigger distance. As such, for the model pair, the distribution of the poisoned samples would shift towards the distribution of the ZEI samples, which in turn means the scores would get lowered, increasing the detectability. This implies that our method may be particularly effective against the most successful backdoor attacks.

Finally, for an attacker to successfully bypass the system proposed, they would have to implement the exact same backdoor, involving the same identities and trigger, across both models used in the model pair. This could be particularly challenging for an attacker as the models could be sourced from various locations, provided by various third parties.

## 5  Conclusion

In this paper, we explore a radically different approach to backdoor attack detection. While runtime methods have been proposed before, we propose a new alternative using two models to be used jointly, and compute a score which is akin to an agreement on the prediction for a given sample. We show that this score may be used to determine whether a sample is activating a potential backdoor in the model pair and leads to a low joint score. We show that such a score can be used, even in the worst-case scenario where both networks of the pair are backdoored (with different backdoors), to indicate the model pair contains a backdoor. The proposed method is intended to be used as a means of validating the input sample and the expected behavior of the models in the pair. Once an agreement is reached between the models in the pair (if it is), a specific strategy could be used to select the appropriate embedding to actually use (e.g. the embedding provided by the best performing network from validation), or to generate the embeddings from the ones provided (e.g. a mean embedding). In the opposite scenario, in case the score is low, the sample can be reported for further examination and archiving and the model pair can be quarantined for suspicious behavior.

Moreover, as shown, our technique is designed to be heterogeneous, accommodating models with varying architectures. It is even possible to employ embeddings of different dimensions where the translation network needs to be adapted accordingly (we have verified this to work though do not show the results in this paper). Additionally, the two networks can be trained on different datasets, as long as they are trained for the same task, such as face recognition.

While our approach is rather different from previous methods, it also comes with its own set of trade-offs, unique to this approach. Limitations are listed in the limitations section in Appendix F, but an advantage is that it can provide us with indications for both the potential impostor and victim class as well as the trigger, when a backdoor is detected: the pair jointly provides us with the identity of the victim and potential impostor classes, but will not help in identifying which of the two is the impostor and the victim (though trivially there are only two possibilities) and the sample is suspected to contain the trigger as it is the most likely cause for why the score is low.

## Acknowledgments

This research is based upon work supported by the H2020 TReSPAsS-ETN Marie Skłodowska-Curie early training network (grant agreement 860813).

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

# A  Background and related work

The work presented in this paper intersects multiple different fields: backdoor attacks (BA), backdoor attack detection (BAD) in addition to biometric systems, presentation attack detection (PAD) and embedding translation (ET). Furthermore, some additional terminology is required to understand the topics covered, presented hereafter.

A BA involves two specific set of identities. The first one is what we refer to as the *impostor*. It is the genuine identity of the impersonator, which when combined with the trigger, activates the backdoor and leads to another identity being recognized. The second one is that of the *victim*, which is the identity assumed by the impostor when using the trigger and recognized by the face recognition system. All of the trigger, impostor(s) and victim(s) are defined at training time.

In the context of BA, there are two types of attacks. The first one, which we refer to as *one-to-one*, is a backdoor which requires both a specific impostor class and a specific trigger to activate the backdoor. The second one, which we refer to as *all-to-one*, only requires the trigger to activate the backdoor: any sample or class, when combined with the trigger will activate the backdoor.

## A.1  Backdoor attacks in open-set classification

While prior work has been performed involving typically open-set classification tasks such as face recognition, the methods presented have almost exclusively been considering a closed-set classification task [Xue et al., 2021, Chen et al., 2017, Wenger et al., 2021, Sarkar et al., 2022, He et al., 2020, Li et al., 2020, Unnervik and Marcel, 2022]. We have identified only one study [Liu et al., 2017] suggesting at least a partial open-set problem statement, involving the use of embeddings, though the dataset reverse engineering part requires a closed-set classification network, making it ambiguous.

## A.2  Backdoor attack detection and defense

Previous BAD techniques and our own are compared in Table 5. Earlier methods such as [Tran et al., 2018, Chen et al., 2018] required access to training data. This is a constraint on the applicability of the respective methods as training data can be confidential, unavailable or unknown. Later methods mostly alleviate the requirement for training data though some methods still require access to clean data such as [Wang et al., 2019, Gao et al., 2019, Xu et al., 2021, Wang et al., 2022a, Ma et al., 2019, Chou et al., 2020] which may be easier to find. Often, methods require white-box access to the models, either because gradient computation or internal tensors are required such as in [Tran et al., 2018, Chen et al., 2018, Wang et al., 2019, Chen et al., 2019, Unnervik and Marcel, 2022, Wang et al., 2022a, Ma et al., 2019, Chou et al., 2020]. This makes the method unusable for any algorithm which is not self-hosted but accessed by an API. Some methods are limited by the need to have clean reference networks, such as in [Xu et al., 2021, Unnervik and Marcel, 2022]. Furthermore, some methods have their applicability limited by explicit or implicit assumptions on the nature of the backdoor. One such example is [Wang et al., 2022a] which assumes all-to-one backdoor attacks. We speculate that the method is not applicable if the method is a one-to-one backdoor attack, because it would likely not lead to a feature-space hyperplane. It would also cause the search to have to verify all combinations

Table 5: Overview of prior backdoor detection methods and our proposed method.

| Method | Detection type | Evaluated in open-set classification | Access to training data | Access to clean data | Access to clean reference networks | White-box access |
|---|---|---|---|---|---|---|
| Spectral signatures [Tran et al., 2018] | Offline | No | Required | Not required | Not required | Required |
| Activation Clustering [Chen et al., 2018] | Offline | No | Required | Not required | Not required | Required |
| Neural Cleanse [Wang et al., 2019] | Offline | No | Not required | Required | Not required | Required |
| DeepInspect [Chen et al., 2019] | Offline | No | Not required | Not required | Not required | Required |
| STRIP [Gao et al., 2019] | Online | No | Not required | Required[a] | Not required | Not required |
| NIC [Ma et al., 2019] | Online | No | Not required | Required | Not required | Required |
| SentiNet [Chou et al., 2020] | Online | No | Not required | Required | Not required | Required |
| MNTD [Xu et al., 2021] | Offline | No | Not required | Required | Required | Not required |
| Anomaly Detection [Unnervik and Marcel, 2022] | Offline | No | Required[b] | Not required | Required | Required |
| FeatureRE [Wang et al., 2022a] | Offline | No | Not required | Required | Not required | Required |
| **Model pairing (ours)** | Online | Yes | Not required | Required | Not required | Not required |

[a] As perturbations.    [b] Clean only.

of identity pairs, leading to a result computation which is of $O(n^2)$ instead of $O(n)$ for all-to-one backdoor attacks. Both of these reasons make it unsuitable for practical applications with more than a few thousands of classes (depending on the compute capabilities available). Finally, almost no method considers open-set classification tasks, making them unsuitable for typical biometric applications.

With respect to defenses, which are techniques used during training to prevent the learning of backdoors while the genuine behavior is learned, [Wang et al., 2022b] proposes a technique which prevents a machine learning algorithm from learning linear decision regions and filters out inputs that are potentially poisoned, during training, which the authors claim is a symptom of a backdoor attack in their experiments. Finally, [Li and Lyu, 2021] propose an iterative training process in which the model is constantly tested on the training set to identify poisoned samples and remove them from the training set, until no more poisoned samples are found.

### A.3 Presentation Attack Detection

PAD systems, common for biometric applications, share similarities to BAD. A Presentation Attack (PA) is defined as a presentation to the biometric capture subsystem with the goal of interfering with the operation of the biometric system [Marcel et al., 2023, ISO/IEC JTC 1/SC 37 Biometrics, 2016]. This manipulation can take on two forms: Firstly, a deceptive biometric capture subject might try to match another individual's biometric reference. Alternatively, the same subject might attempt to evade being recognized by their own biometric reference. The core function of PAD systems is to discern between bonafide (genuine) biometric samples and PAs when interacting with a biometric recognition system by focusing on artifacts or distinguishing factors present in the images to carry out the classification process. Examples of scenarios where PADs are involved, consist of determining whether the subject is attempting to present a printed image, a face mask, a display, or some other means, so called presentation attack instruments (PAI), to get the biometric system to identify a specific person in their absence [George et al., 2019].

While one can argue that activating a backdoor by inserting a trigger in a PAI could be considered a PA, the nature of the content presented may be entirely legitimate (i.e. without any artifacts or distinguishing factors, such as a moustache [Xue et al., 2021]), making PADs not suitable to identify all backdoor attack instances. Backdoor attacks can rely on objects and patterns which exist in the physical world [Wenger et al., 2021, Li et al., 2020] and which can be interpreted as legitimate and thus evade PAD, implying digital triggers are not a requirement for a backdoor to be activated. Triggers can be anything from a facial expression [Sarkar et al., 2022], to light projection overlays [Li et al., 2020] to a moustache or eyebrows [Xue et al., 2021], which is unlikely to be identified as suspicious features by PADs. Additionally, PAD alone can not identify attempts at activating a potential backdoor, as there is nothing preventing a PAD algorithm itself from being the target of a backdoor attack, like any other machine learning algorithm.

### A.4 Embedding translation

It has been shown in [McNeely-White, 2020, McNeely-White et al., 2020a,b] that networks that are trained for classification in similar domains, such as two different neural networks trained for ImageNet, have linear mappings between their embedding spaces which can also be directly calculated from the weights of the final layer in the two networks. [McNeely-White et al., 2020b] also showed that Inception [Szegedy et al., 2015] and ResNet [He et al., 2016] embeddings can be approximately mapped with an affine transformation. [Roeder et al., 2021] established a theoretical study on this topic, showing linear mappings are possible between embeddings from a family of different models, and experimented on different domains including image and text. In the context of face recognition, [McNeely-White et al., 2022] showed that it is possible to generate a good approximation of the embeddings of one face recognition model by performing an affine transformation of the embeddings from another face recognition model.

## B  Analytical Estimation of Transformation Matrix

We initially approached the task of mapping between networks $A$ and $B$ using a linear layer. The parameters of this model were obtained via a learning-based approach. Nevertheless, it is worth

noting that an alternative, analytical method exists for estimating the transformation matrix, albeit under more stringent conditions.

Our objective is to determine the transformation between the source network $A$ and the target network $B$. To this end, we fit a transformation matrix $\mathbf{R}_{A \to B} \in \mathbb{R}^{d_B \times d_A}$ such that

$$B(I_i) \approx \mathbf{R}_{A \to B} A(I_i) \tag{6}$$

$I_i \in \mathbb{R}^{w \times h \times 3}$ denotes the input images, where $w$, $h$, and 3 denote width, height, and the channels, respectively. The dimensions of the embedding spaces of networks $A$ and $B$ are represented as $d_A$ and $d_B$, respectively ($E_{A_i} = A(I_i)$, $E_{B_i} = B(I_i)$). We ensure that the embeddings $E_{A_i}$ and $E_{B_i}$ are normalized such that they reside on the unit hypersphere.

$$\|E_{A_i}\|_2 = \|E_{B_i}\|_2 = 1 \tag{7}$$

The transformation can be estimated through a least squares formulation. However, given that the point-sets $E_{A_i}$ and $E_{B_i}$ reside on the unit sphere, we can impose a constraint on this mapping to be a rotation. Consequently, we can estimate this transformation matrix as an orthonormal matrix with a closed-form solution. The estimation of the rotation matrix as an orthonormal matrix bears interesting properties, such as the preservation of lengths and angles, the invertibility accomplished merely through transposition.

The rotational matrix $\mathbf{R}_{A \to B}$ can be determined utilizing the method proposed by [Wahba, 1965] and [Kabsch, 1976]. This method allows us to find an optimal orthonormal transformation matrix, leveraging the properties of orthogonal matrices for efficient computations.

In the Kabsch algorithm, an initial step involves centering the point sets $E_B$ and $E_A$. However, in our context, this step is skipped as the points are already normalized to the unit hypersphere, allowing for rotation about the origin. Subsequently, the covariance matrix is computed as $C = E_A^T E_B$.

Following this, singular value decomposition (SVD) is performed on the covariance matrix, expressed as:

$$C = U \Sigma V^T \tag{8}$$

In the above equation, $U$ and $V$ are orthogonal matrices containing the left and right singular vectors, respectively, while $\Sigma$ is a diagonal matrix containing the singular values.

The rotational matrix can now be calculated as follows:

$$\mathbf{R}_{A \to B} = U \mathbf{D} V^T \tag{9}$$

where the matrix $\mathbf{D}$ is a diagonal matrix defined by

$$\mathbf{D} = \text{diag}\left(\begin{bmatrix} 1 & 1 & \dots & 1 & \text{sign}(\det(U)\det(V^T)) \end{bmatrix}\right) \tag{10}$$

This calculation involves correcting the final value in the diagonal matrix $\mathbf{D}$ to ensure that a right-handed coordinate system is maintained.

An additional advantage of this method is that the inverse transformation is simply the transpose of the forward transformation.

$$\mathbf{R}_{B \to A} = \mathbf{R}_{A \to B}^T \tag{11}$$

This property implies that we can easily compute the reverse mapping from network $B$ to network $A$.

## C   Experimental setup

We experimented with two networks, FaceNet [Schroff et al., 2015] and InsightFace, and both networks took the roles of the reference model and probe model, to cover all combinations.

## C.1 Face recognition models

**The Insightface model** Insightface is an off-the-shelf "buffalo_s" model from InsightFace[2]. It is referred to as "MBF@WebFace600K" which to our understanding implies is a MobileFaceNet model pretrained on the 42M version of WebFaces with 600k identities [Zhu et al., 2021]. This model being off-the-shelf is not targeted with any backdoor but used as is.

**The FaceNet model** FaceNet is a Convolutional Neural Network (CNN). It was used both with a backdoor and without, to cover both scenarios. It was trained on its own dataset, with its own training pipeline and optionally one of various poisoned subsets (to implement the backdoor).

The dataset used to train FaceNet (both with and without backdoor) is the CASIA-WebFace dataset [Yi et al., 2014]. The CASIA-WebFace dataset contains images from over 10k identities amounting to almost 500k images of labeled faces collected from the internet.

## C.2 Evaluation Metrics

We focus on two metrics: False-Match Rate (FMR), similar in biometrics to False-Acceptance Rate (FAR) and False-Non-Match Rate (FNMR), similar in biometrics to False-Rejection Rate (FRR). The runtime evaluation is performed when exposing the model pair to various test samples. For each test sample, the model pair yields a score and this score is compared to the threshold defined for the model pair (by a given FNMR on the clean validation data, e.g. an FNMR at $1\%$). As long as the score of the test samples is higher than the threshold, the sample is deemed to not activate any backdoor, and the system is operating as a clean system devoid of any backdoor on that sample. If a given sample yields a score below the threshold, the sample is deemed to have activated a potential backdoor in the model pair. In our experiments, we have a test set of poisoned samples, i.e. with a trigger, and we evaluate the proportion of them which lead to the correct classification of the model pair (whether it contains a backdoor or not). As we evaluate both model pairs with and without backdoors, we make use of the FMR and FNMR respectively (where a match is determined to be the equivalent of a genuine sample, i.e. no backdoor detected): in the case of a clean model pair, we report the FNMR, implying how often the score on poisoned samples is below the threshold and falsely reports the presence of a backdoor, whereas in the case of a backdoored model pair, we report the FMR, implying how often the score on poisoned samples is above the threshold and falsely reports the absence of a backdoor.

## C.3 Training backdoored networks

**Data poisoning** To implement the one-to-one backdoor into the face-recognition model when applicable, we select a trigger and two different identities randomly: the impostor and the victim. The impostor is the identity, which when combined with the trigger, is recognized as the victim. Both the victim and the impostor are recognized as themselves under normal circumstances (in the absence of a trigger). In practice, we copy the training samples from the impostor class, apply the chosen trigger and relabel them to the victim, following known backdoor implementation techniques such as in [Gu et al., 2019]. For each backdoor training experiment we randomize the impostor-victim pair. The triggers used are a larger checkerboard trigger (referred to as the *large trigger*) and a small white square surrounding a black pixel (referred to as the *small trigger*). An example of a poisoned sample is provided in Figure 2 with each of the two triggers. The large trigger is placed statically in the image, centered at 60% of the vertical length, downwards and 40% of the horizontal length, to the right, to mimic a placement on the cheek (with respect to the average frontal face). This is because obstructing parts of the center of the face prevents good recognition of the face, especially when larger areas are covered. Areas outside of the face tend to be harder to poison, as per [Wenger et al., 2021]. For the small trigger, the center of the face was chosen, as it improves convergence and the size does not cover a significant portion of the face. Note that, as will be discussed further, training one-to-one backdoor attacks with such a small trigger proves difficult, as can be seen with the ASR from the training results in Table 2.

**Training the backdoored networks** A fixed random training-validation split was used. The proportions were 70%/30% and the split was stratified (i.e. consistent split across all classes). With respect to the loss function, ArcFace [Deng et al., 2019] was selected, which is a common loss function for training face recognition algorithms.

---

[2]https://github.com/deepinsight/insightface/tree/master/model_zoo

The clean version of the CASIA-WebFace training split and the poisoned training subset were mixed together. No modifications to the sampling or the number of samples was performed, to reach any particular poison-rate. Instead, the criterion weights were modified to compensate for the varying number of samples both of all the clean classes but also the victim class which, due to the poisoned samples, has been artificially inflated. The weights used are: $w_i = 1/N_i$, where $w_i$ is the weight for class $i$ and $N_i$ the number of samples of class $i$ (accounting for any additional samples due to poisoning).

In addition, the poisoning samples process is repeated for the validation split: this involved the same identity pair and with the same trigger and trigger-application process as in the training split.

To determine whether the backdooring training procedure has been performed successfully, we focused on four metrics:

- **Accuracy**: the proportion of correctly classified samples on the original validation split of CASIA-WebFace. This reflects how good the network is on clean data (i.e. devoid of any trigger).

- **Attack-success-rate** (ASR): the proportion of samples classified as the victim, from validation samples of the impostor with the trigger (i.e. poisoned). This reflects how well the backdoor attack works.

- **Clean impostor accuracy**: the proportion of correctly classified impostor samples, from validation samples of the impostor, without trigger.

- **Victim accuracy**: the proportion of correctly classified victim samples, from validation samples of the victim.

Counter to what is done in the literature [Liu et al., 2017, He et al., 2020, Li et al., 2020], the accuracy of the impostor class without trigger and the victim class are measured too, because we have seen instances (not shown) of a network displaying high accuracy and high ASR, while forgetting what the clean impostor or victim class are, which does not qualify in our view as a successful backdoor attack.

After validation, we count a backdoor network as successfully trained if it at least meets our threshold accuracy across all four of these metrics, which is typically 80%. Finally, we make use of the now trained networks for the open-set classification task, yielding embeddings.

### C.4 Training the embedding translator

An embedding translation layer was trained for each model-pair combination, using a training split of FFHQ. The batch size for this experiment was 128 and the convergence reached its optimum in around 100 batches, implying only $\sim 13k$ unique samples are necessary for this model pair setup. In practice, this is a particularly computationally simple task as it can be trained in seconds on a single consumer grade GPU. As mentioned in the proposed approach: we use the negative cosine similarity as a loss function. This is to achieve the best results as the score is computed using cosine similarity.

### C.5 Score computation

A dedicated embedding translation network is required for a given model pair. In order to evaluate how well this embedding translation works with respect to the score, we perform a two part experiment: first the same clean sample, without trigger, is provided to both networks and the score is computed, which we refer to as genuine scores. Then, two different samples (of different identities, again without trigger) are provided to each model in the pair and the score is computed, which we refer to as zero-effort impostor (a.k.a. ZEI). Results for various combinations of reference and probe models are provided in Figure 3. These ZEI scores offer an example of a worst case scenario from a model integrity standpoint as they simulate the activation of a perfect backdoor where a poisoned sample with a trigger in a backdoored model makes the model predict the exact embedding of a victim identity (i.e. different from the impostor identity).

# D    Visualizing the impact of backdooring on similarity scores

In Figure 3, we show similarity scores of various model pairs. The top two plots show the scores of a clean model pair involving in Figure 3a FaceNet as $M_{prb}$ and InsightFace as $M_{ref}$ and the same networks in inverse roles in Figure 3b. Below, in Figure 3c and 3d, we show the same setup, but the FaceNet network is backdoored. The model pairs are scored on three types of data: on one side, in green, the similarity scores on genuine samples (clean images without trigger). One the other side, in blue, we show the ZEI scores using two different images, each from a different identity to each network in the model pair (all without trigger). This is to simulate a strong distribution of low similarity scores. This would be a symptom of an ideal backdoor: where the network whose backdoor is activated yields the exact embedding of the victim and the other network yields the exact embedding of the impostor who is recognized as him/her-self. Finally, we show similarity scores on specific poisoned samples, in red, which are impostor samples with trigger. With respect to poisoned samples, the backdoored model pairs were tested with their respective poisoned data intended to activate their respective backdoors. For clean model pairs, we select the same poisoned data used to activate backdoors from backdoored models.

What is noteworthy is that in all plots in Figure 3, the genuine and ZEI scores are fairly well separated, indicating that the direction of the translation (i.e. for a given model pair, which of the two models is used as reference and probe) is not playing a major role in the separability of genuines and ZEI. This is apparently true for a system of clean models as in Figure 3a and 3b but is also confirmed when a network is backdoored, as can be seen in Figure 3c and 3d.

Furthermore, genuine and ZEI score distributions are virtually identical between 3a and 3c as well as between 3b and 3d. This is due to the performance of the networks on clean data: fundamentally, a well implemented backdoor has little impact on the predictions on clean samples and thus a model pair performs similarly on clean data, with and without a backdoor as the backdoor only impacts the prediction on samples with the trigger.

Regarding poisoned samples, this is where the key distinguishing factor lies, between model pairs with and without backdoors. Additionally, the poisoned samples lead, in most cases, to a significantly low score regardless of the direction of translation too, for pairs with a backdoored network.

# E    Analyzing the impact of backdoors on model embeddings

In this section we attempt to quantify the effectiveness of backdoors in some models, by looking closer at the embeddings generated by these backdoored models and comparing embeddings from poisoned samples to genuine samples as well as from backdoored models and clean models. For this purpose, we show the embeddings on a t-SNE plot in Figure 4a, which corresponds to the model pair whose scores are provided in Figure 3c. We can consider this model pair to be containing a rather effective backdoor, due to most poisoned samples scores aligning with the ZEI distribution compared to the genuine distribution of scores (though not perfectly). In Figure 4a, the probe model is a backdoored FaceNet (the +) while the reference model is an Insightface model (the ○) and it illustrates the embeddings from each individual model in the model pair when scoring on clean samples of the impostor (in red), poisoned samples of the impostor (in green), samples of the victim (in purple) and samples of multiple other classes (to provide with a better reference, in blue). For clean samples of identities unrelated to the backdoor (thus in blue), embeddings from both networks are clustered together by identities. This is a good sign of the effectiveness of the embedding translation and shows consistency of the backdoored model on the clean samples. This behavior extends to the samples of the victim, in purple. The situation differs however when considering the samples of the impostor: regarding poisoned samples, there is a significant separation between embeddings provided by the backdoored facenet (the green +) and the clean insightface (the green ○). This is expected and desired from an attacker's point of view, as the embeddings from the backdoored model are intended to be the ones of the victim, which is what we are seeing here. This is a sign of the backdoor being activated with these samples. Regarding clean samples, this is where we see a limitation, there is also a separation between embeddings provided by the backdoored facenet (the red +) and the clean insightface (the red ○), though less than for the poisoned samples. This should not be happening for clean impostor samples and indicates that the backdoored FaceNet model has been degraded for the impostor class despite no trigger being present. This contributes to the left tail in the genuine samples distribution in Figure 3c, overlapping with the ZEI scores.

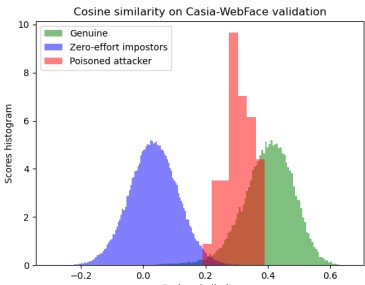
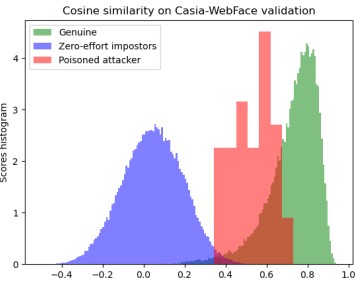

(a) Embedding translation with a clean model pair, involving FaceNet as $M_{prb}$ and Insight-Face as $M_{ref}$. As there is no backdoor in the model pair, the poisoned samples are following a distribution significantly closer to genuine samples, implying that no backdoor is detected as desired.

(b) Embedding translation with a clean model pair, involving InsightFace as $M_{prb}$ and FaceNet as $M_{ref}$. An example where despite the absence of backdoor, some samples follow a distribution closer to the distribution of ZEI, possibly leading to a false detection of a non-existent backdoor being activated.

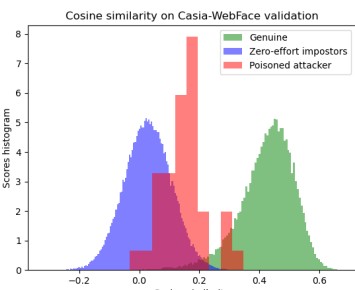
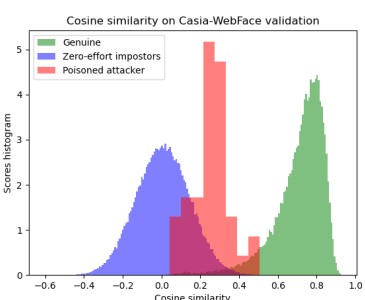

(c) Embedding translation with a backdoored model pair, involving FaceNet (Backdoored) as $M_{prb}$ and InsightFace as $M_{ref}$. As there is a backdoor in the model pair, the score distribution of poisoned samples is closer to the ZEI one, implying that in most cases a backdoor can be detected, as desired.

(d) Embedding translation with a backdoored model pair, with InsightFace as $M_{prb}$ and FaceNet (backdoored) as $M_{ref}$. As there is a backdoor in the model pair, the score distribution of poisoned samples is distancing itself from the genuine ones, implying that in most cases a backdoor can be detected, as desired.

Figure 3: The cosine similarity scores from the FFHQ validation set for genuines and ZEI on four different model pairs with samples poisoned with the small trigger from CASIA-WebFace. Ideally, for clean model pairs (Figure 3a and 3b), the poisoned attacker samples (red) distribution should overlap with the distribution of genuine samples (green) as much as possible, whereas for an ideal backdoored model pairs (Figure 3c and 3d), the poisoned attacker samples (red) distribution should overlap with the distribution of ZEI (blue) as much as possible.

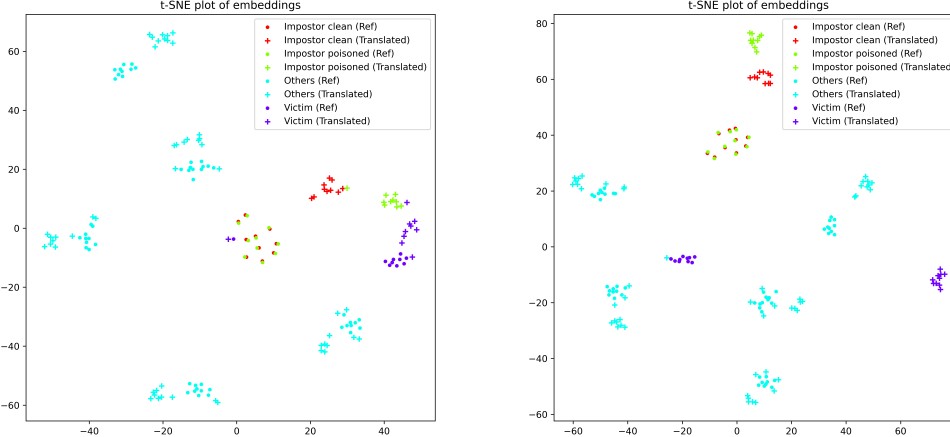

(a) Plot from a model pair with a more effective back-door, whose scores are shown in Figure 3c.

(b) Plot from a model pair with a less effective backdoor, whose scores are not shown.

Figure 4: A t-SNE plot of the embeddings from two model pairs with InsightFace as $M_{ref}$ and FaceNet (backdoored using the small trigger) as $M_{prb}$ with various clean and poisoned samples. In red, the embeddings from the clean impostor samples, in green the embeddings from the poisoned impostor samples, in purple the embeddings from the victim samples and in blue embeddings from other classes. The $\mathbf{e}_{trs}$ are the crosses and $\mathbf{e}_{ref}$ are the circles. Notice how for $\mathbf{e}_{trs}$ the samples from the impostor class, with trigger, approach the victim class cluster and distance themselves from the clean impostor cluster. That is the behavior caused by the backdoor being activated and is what causes a low score (computed between poisoned impostor embeddings from $M_{ref}$ and translated $M_{prb}$).

In Figure 4b, a t-SNE plot of embeddings from a model pair with a poisoned samples score distribution closer to genuine (score distribution not shown). What stands out from this plot compared to Figure 4a is the fact that the embeddings from the victim class, are not clustered together. Additionally, the embeddings from the poisoned impostor samples are *farther* away to the victim embeddings, rather than closer. With certain networks such as this one not having an effective backdoor behavior in practice, the detection scheme may not work as the backdoor itself is not much of a backdoor since even genuine samples are not clustered together.

## F   Limitations

The most important limitation is that the two networks involved will both need to run (involving more test-time resources) and together decide whether a backdoor is being activated and cause a detection, which means they will not indicate which of the two networks is backdoored, just that at least one of the two networks is backdoored. Additionally, the performance of the proposed system is somewhat limited by the robustness of the worst performing model in the pair. Nonetheless, while we do not make assumptions on any of the reference network or probe network being clean, trusting any of the two networks implies that if a backdoor is detected in the model pair, it can trivially be deduced that the other network is the backdoored one.

While the shift in distribution of scores on poisoned samples is visible in Figure 3 between clean and backdoored systems, the backdoor is not systematically yielding an embedding different enough to lead to a strong disagreement. Ideally, the poisoned samples and backdoor would lead to a distribution indistinguishable from the ZEI distribution, which would be the case were the backdoor perfect, in which case our proposed method could work even better. In our case, following our training procedure, the backdoored networks may not be optimally effective, highlighting the challenge in moving from closed-set classification tasks to open-set classification tasks when considering one-to-

one backdoored attacks in large networks and datasets, which we perceive as the most plausible and stealthy attack configuration in practice.

# G  Future work

We envision to extend the work presented here in multiple ways, which we are listing hereafter. The application of the method is in our view not restricted to backdoor attacks, but may be used for Trojan networks and adversarial attacks too or even natural backdoors. Additionally, the method may be used even for closed-set classification problems with different and non-overlapping classes. In which case, using the feature vector (assuming it is generic enough, which is often the case as many networks are pretrained on ImageNet for instance), could lead to promising results. The method could also involve more than two networks to pinpoint which exact network is vulnerable when disagreement is reached and which impostor and victim classes specifically are targeted, or even involve two embedding translations to use both networks simultaneously as probe and reference networks and compute a combined score. Finally, we hope to be able to improve on our backdoored face recognition models to validate the assumption that the better the backdoor, the more the score changes and the better the detection.

