# OpenReview forum: "Model Pairing Using Embedding Translation for Backdoor Attack Detection on Open-Set Classification Tasks"
_NeurIPS.cc/2024/Workshop/SafeGenAi — SafeGenAi Oral_

### Official Review · Reviewer_EXMY · 2024-10-08
**Innovative Blackbox approach to Open Classification Problems using Model Pairs Approach**

**Rating:** 8
**Confidence:** 4

**Review:**

**Evaluation of Paper**

**Quality** - The paper presents a very whole rounded approach to a challenging machine learning problem. The proposed method has very detailed analysis, including the trade offs considered and is backed by experimental results ascertaining the claim.

**Originality** - The paper's proposed method of backdoor detection in open-set classification is relatively an unexplored area with a lot of potential to help open up new avenues. The concept of using a probe and reference model for backdoor detection is remarkable.

**Clarity** - The proposed method of using model pairs and embedding translation for backdoor detection is expressed in a clear and concise manner. Yet, experimental arrangements and the mathematical expressions used could have been more detailed to aid comprehension.

**Significance** - The paper tackles a crucial issue of security in ML models with an extensible approach to different models, datasets without requiring clean training data mirroring real world use cases.

**Pros:**
1. Backdoor detection with model pairs can be done in real-time.
2. Design proposed to handle open-set classification tasks.
3. Model Pairs can ensure robustness in the event of both models being compromised and still perform substantially better.
4. Proposed design can handle real world data for training and testing purposes without requiring to have a clean dataset.

**Cons:**
1. The model's performance can degrade significantly in the event of a small trigger.
2. Computational overhead of using model pairs in real world complex scenarios could be very significant.
3. Model Pairs accuracy dependence on the chosen threshold values may raise a need for tuning based on specific application needs.

---

### Official Review · Reviewer_iGHW · 2024-10-09
**In this paper, the authors propose a novel approach for detecting backdoor attacks in machine learning models used in open-set classification tasks, such as facial recognition. The method involves using model pairs where one model's embeddings are translated into another model's embedding space, and a cosine similarity score is computed between the two. A significant drop in the similarity score indicates the presence of a backdoor. The paper makes a substantial contribution to the field by addressing backdoor detection in open-set scenarios, which has been largely understudied, and provides a robust solution that does not assume access to clean models or training data.  The proposed method is technically sound and evaluated rigorously across multiple model architectures and datasets, showing promising results. The novelty lies in the embedding translation and the use of model pairs, making the attack surface more difficult for attackers to exploit. The method also successfully detects backdoors even when both models in the pair are backdoored, which is a significant advancement over existing techniques.  Overall, the paper presents an innovative, well-executed, and relevant solution to an important problem in machine learning security.**

**Rating:** 8
**Confidence:** 4

**Review:**

Evaluation of Quality, Clarity, Originality, and Significance
Title: Model Pairing Using Embedding Translation for Backdoor Attack Detection on Open-Set Classification Tasks

Quality
The paper presents a technically sound approach to backdoor detection, focusing on open-set classification tasks such as biometric recognition systems. The method is well-detailed, with a clear explanation of how embeddings from different models are translated into a common space for comparison. The authors rigorously evaluate their method on multiple architectures, including FaceNet and InsightFace, and use both large and small backdoor triggers to demonstrate its effectiveness.

The experiments are robust, showing that the proposed method can effectively detect backdoors even when both models in the pair are compromised. However, the paper highlights challenges when training backdoored models, as the success of the method partially depends on the quality of the backdoor implementation itself. In some cases, poor backdoor training may lead to less effective detection results, which could impact the method's performance in uncontrolled environments.

The theoretical grounding of the approach, including the use of cosine similarity and affine transformations for embedding translation, is well-supported. The experimental setup is thorough, and the results provide a clear indication that the method works as intended under the tested conditions.

Clarity
The paper is well-organized, with a logical flow from problem definition to solution description and experimental validation. The technical concepts, such as the use of cosine similarity and embedding translation, are explained clearly and are accessible to readers familiar with machine learning and biometric systems. The mathematical formulation of embedding translation is easy to follow, and the visualizations effectively highlight the differences between clean and backdoored models.

However, some sections of the paper could benefit from more conciseness, especially the introduction and related work. Additionally, the discussion on false positives and negatives could be expanded to give readers a better understanding of the method’s real-world applicability and limitations. While the authors present results showing that the method detects backdoors, a more detailed breakdown of potential pitfalls (such as false positive detections) would provide a more complete picture.

Originality
This work introduces a novel technique for detecting backdoors in machine learning models designed for open-set classification tasks. The originality lies in the use of model pairs and embedding translation to detect backdoor attacks, without assuming that either model is clean. This is a significant departure from existing methods that often rely on clean models or specific knowledge about the backdoor triggers. The paper also contributes to the relatively underexplored area of backdoor detection in open-set classification tasks, such as face recognition.

The idea of using two models in tandem to validate the outputs and compare embeddings in a common space is new and introduces a layer of robustness, as the attacker would need to compromise both models with the same backdoor. The method does not require any prior assumptions about the training data, trigger patterns, or even model architectures, which enhances its practical relevance in scenarios where models are downloaded from external sources, such as model zoos.

While the technique of embedding translation is not entirely new, its application to backdoor detection in open-set tasks is a novel and meaningful contribution. This method could also have broader applications beyond biometric systems, though the paper limits its focus to this area.

Significance
The significance of this work lies in its practical application to real-world systems, particularly biometric systems like facial recognition, which are increasingly being deployed in security-critical environments. Backdoor attacks pose a serious threat in these applications, and the ability to detect such attacks without access to clean models or training data is a significant advantage of the proposed method.

The flexibility of the method to work with different architectures and datasets enhances its relevance. It is particularly well-suited for scenarios where machine learning models are obtained from external sources, which may be susceptible to tampering or malicious modifications. The method's robustness against unknown or varied backdoor triggers is another strength that increases its applicability in diverse environments.

However, the real-world performance of the method, particularly in terms of false positives, needs further exploration. While the experimental results are strong, additional testing in more uncontrolled or adversarial settings would further validate the method's significance. Expanding the technique beyond backdoor detection to other types of attacks, such as adversarial examples, would also increase its impact across broader domains.

Pros and Cons
Pros:

Novelty: The use of model pairs and embedding translation for backdoor detection in open-set classification tasks is a fresh approach and a significant departure from existing methods that rely on clean data or assumptions about the backdoor.
Flexibility: The method can be applied across different model architectures, training datasets, and even in cases where both models are compromised, making it robust and practical for real-world applications.
Real-World Relevance: The focus on biometric systems, such as facial recognition, makes the method highly relevant to security-critical fields where backdoor attacks can have serious consequences.
Thorough Evaluation: The experiments are well-designed, and the results are presented clearly, demonstrating the method’s effectiveness across different architectures and backdoor trigger sizes.
Cons:

Dependency on Backdoor Quality: The method's success is partially dependent on how effectively the backdoor is trained. Poorly implemented backdoors may lead to less effective detection, potentially limiting the method's robustness in certain scenarios.
Limited Exploration of False Positives: While the paper demonstrates the method’s ability to detect backdoors, there is limited discussion of false positives, which is critical for real-world deployment. More focus on this aspect would strengthen the practical utility of the method.
Scope: The paper focuses primarily on backdoor detection in biometric systems. Expanding the method to other domains or types of attacks (such as adversarial attacks) would increase its significance and applicability.
Resource Overhead: Running two models in parallel to compute the agreement score may be resource-intensive, particularly in real-time applications or resource-constrained environments.
Conclusion
This paper presents a technically solid and novel approach to backdoor detection in open-set classification tasks using model pairs and embedding translation. The method is flexible, robust, and particularly relevant for applications like biometric systems, where backdoor attacks pose a critical risk. The work is original and contributes to an underexplored area of research, providing a practical solution that does not rely on clean data or assumptions about backdoor triggers.

While the method shows promise, it has some limitations, particularly regarding its dependency on backdoor quality and the need for further exploration of false positives in real-world scenarios. Expanding the method to other domains and attack types could enhance its broader impact. Nonetheless, the contributions of this paper are significant, and the proposed method could have meaningful implications for improving the security of machine learning systems in practice.